# Coping Strategies, Anxiety and Depression in OCD and Schizophrenia: Changes during COVID-19

**DOI:** 10.3390/brainsci11070926

**Published:** 2021-07-13

**Authors:** Ángel Rosa-Alcázar, José Luis Parada-Navas, María Dolores García-Hernández, Sergio Martínez-Murillo, Pablo J. Olivares-Olivares, Ana I. Rosa-Alcázar

**Affiliations:** 1Department of Psychology, Catholic University of Murcia, 30107 Murcia, Spain; aralcazar@ucam.edu; 2Facultad de Educación, Universidad de Murcia, 30107 Murcia, Spain; jlpn@um.es; 3Department of Personality, Assessment & Psychological Treatment, University of Murcia, 30100 Murcia, Spain; mariadogh@um.es (M.D.G.-H.); sergio.martinez21@um.es (S.M.-M.); pjoo@um.es (P.J.O.-O.)

**Keywords:** obsessive–compulsive disorder, schizophrenia, coping strategies, anxiety, depression, COVID-19

## Abstract

Background: The main aim was to examine changes in coping strategies, anxiety and depression in obsessive–compulsive and schizophrenia patients during COVID-19, in addition to controlling the influence of intolerance to uncertainty and experiential avoidance. Method: The first time, the study comprised (15–30 April 2020) 293 patients, 113 of whom were diagnosed with obsessive–compulsive disorder, 61 with schizophrenia and 119 healthy controls, aged 13–77 years (*M* = 37.89, *SD* = 12.65). The second time (15–30 November), the study comprised 195 participants (85 obsessive–compulsive patients, 42 schizophrenic patiens and 77 healthy controls participants). The evaluation was carried out through an online survey. Results: The clinical groups worsened over time in cognitive coping, anxiety and depression, while the control group only worsened in depression. Intergroup differences in anxiety, depression and coping strategies were maintained, highlighting the use of some maladaptive strategies (avoidance, spiritual) in clinical groups. Experiential avoidance and tolerance for uncertainty mainly affected coping. Conclusions: The duration of COVID-19 not only produced changes in anxiety and depression in clinical groups but also in coping strategies to face this pandemic and its consequences.

## 1. Introduction

Since the beginning of the coronavirus pandemic, the population has been subjected to an anxiogenic stimulus for an unusually long period of time. Previous studies have reported that controllable or uncontrollable stressful experiences may provoke or exacerbate psychological and physical psychopathologies. It is important to take variables into account both of the stressor itself—frequency, duration, type, number, anger, ability to control—and own individual—age, coping skills, fear level, tolerance for uncertainty, previous psychopathologies, etc. [1,2].

The duration of COVID-19 is a factor associated with increased emotional distress. Specifically, the Chinese population showed an increase in the level of depression from the first to the second wave. Age, income and coping strategies were variables influencing hostility and level of perceived stress [3].

Prevention behaviors (social distancing, hand washing, use of a mask, isolation, among others) can increase the risk of presenting mental illnesses such as anxiety disorders, post-traumatic stress disorder, obsessive compulsive disorder, etc. [4,5,6,7].

Other individual variables, such as the level of fear and social isolation, also play an important role in terms of general well-being and emotional stability, the consequences being greater in people with previous psychopathologies, such as obsessive–compulsive disorder, anxiety and psychosis [8]. Focusing on OCD, the elements of containment of the pandemic will produce reinforcement of obsessive thinking and compulsive behavior and exacerbation of symptoms [7,9,10]. Other authors reported a higher perception of risk, fear, anxiety and stress in a schizophrenic population during a swine fever pandemic [8].

Social isolation is a risk factor for the mental disorders, being a mediating factor for depression, anxiety, suicidal ideation and psychotic episodes [11,12].

Tolerance of uncertainty is another variable related to mental health. This can be defined as the dispositional capacity to withstand the aversive response caused by the perceived absence of key or sufficient information, being sustained by the associated perception of uncertainty [13]. Rettie and Daniels [14] reported that levels of anxiety and depression during COVID-19 were higher in people with greater intolerance of uncertainty and who presented maladaptive coping strategies. Wheaton et al. [15] observed a positive relationship between obsessive–compulsive responses, health anxiety and fear during the pandemic and intolerance of uncertainty. This variable has also been associated with paranoid ideation, conspiratorial thinking and anxiety [16].

Experiential avoidance has been linked to emotional distress. This is understood as a psychopathological process encompassing cognitive, emotional and behavioral avoidance. People with high scores in this variable avoid negative thoughts, feelings and situations to reduce anguish that this generates. However, such avoidance increases the frequency of anxiety-provoking thoughts, leading to greater distress. Some authors [17] have reported that experiential avoidance could transform fear of COVID-19 into obsessive–compulsive symptoms, while others [18] have observed an increase in experiential avoidance and symptoms of anxiety/depression in the obsessive–compulsive population.

Adaptive coping strategies are a protective health factor. Lazarus and Folkman [19] defined these as cognitive and behavioral efforts in order to respond to situations that they consider exceed a person’s resources, as a result of trying to change those situations and regulate emotional reactions [20]. Rosa-Alcázar et al. [21] reported that patients with obsessive–compulsive disorder (OCD), during the first wave of the pandemic, presented higher means than healthy participants in instrumental support, religion, denial and self-blame, being related to anxiety and depression.

Sex, age and educational level have been predictors or mediators of mental health during COVID-19 [3]. Similarly, having suffered from the psychopathological disorder or having relatives with an infected member also influenced an increase in stress, anxiety and depression [22].

Taking into account the progression and duration of COVID-19, the first objective of this study was to examine the temporal change in coping strategies, levels of anxiety and depression in two groups of patients with previous psychopathologies OCD and schizophrenia (SCZ) versus a healthy control group (CG). The second aim was to analyze whether coping strategies could be influenced by experiential avoidance and intolerance to uncertainty. The third objective sought to verify whether sex, age, educational level and whether or not having suffered from COVID-19 could influence the use of one type of coping or another in a group.

## 2. Materials and Methods

### 2.1. Participants

At first evaluation (T1, April 15–30, 2020), there were 293 participants aged between 13–77 (*M* = 37.89, *SD* = 12.65), 113 OCD, 61 SCZ (*n* = 61) and 119 CG, 55.2% women. At the second evaluation (T2, 15–30 November 2020), there were 85 obsessive–compulsive patients, 42 schizophrenic patiens and 77 healthy control participants. The clinical participants were medicated with antidepressants and, in some cases, antipsychotics.

Inclusion criteria for participants were: (i) OCD and SCZ participants included a diagnosis according to DSM (CIE criteria. (ii) The control group (CG) could not present any current psychopathological disorder at that time or in the 6 months prior to the study. The exclusion criteria in clinical groups were: (i) presenting other clinical problems as the main diagnosis and (ii) SCZ participants being in the active phase of the disease and not understanding the questions in the questionnaires.

Sample characteristics in April are presented in Table 1.

### 2.2. Procedure

The study met the ethical standards of the Declaration of Helsinki and has been approved by the Ethics Committee of the University of Murcia, Spain (ID: 2123/2018, Spain). All participants or their tutors provide written informed consent.

The procedure was as follows: (a) Contact some OCD and SCZ associations/public and private clinics/university workers of Murcia. The link was only sent to clinical participants diagnosed with these disorders. (b) Complete the anonymous online survey in 20 min. Responses were saved on a secured server at the University of Murcia. The test presentation order was the same for all participants. Participation was voluntary and free. Evaluation times were 15–30 April and 15–30 November 2020. The CG was selected to be equivalent in age and sex to the OCD group (larger sample size). The recruitment process is shown in Figure 1.

### 2.3. Measures

Spanish Version of the Brief COPE-COPE-28 [20]. Twenty-eight items, 14 subscales and 4 second-order factors answered on a Likert-type ordinal scale of 4 response alternatives (from 0 to 3). Second-order factors are: cognitive coping, social coping, avoidance coping and spiritual coping. Higher scores reflect a higher tendency to implement the corresponding coping strategies. We asked participants to report how often they used the strategy described in each project to respond to COVID-19. Cronbach’s alpha at April and November was 0.80 and 0.79.

Hospital Anxiety and Depression Scale (HADS) [23]. Self-report measure of anxiety and depression of 14 items rated on a 4-point Likert scale (0 to 3). It was divided into anxiety (HADS-A) and depression (HADS-D) subscales both containing seven items. Cronbach’s alpha in this study was 0.77 and 0.79 (depression non-clinical and clinical groups), 0.75 and 0.77 (anxiety non-clinical and clinical groups) and 0.82 and 0.84 (total non-clinical and clinical groups).

Brief Experiential Avoidance Questionnaire (BEAQ) [24]. Self-report measure of experiential evasion of 15 items rated on a 6-point Likert scale (1 to 6). Higher scores indicate greater experiential avoidance. Cronbach’s alpha in this study was 0.90 (CG), 0.87 (OCD) and 0.91 (SCZ).

Intolerance of Uncertainty Scale (IUS) [25]. Comprising 27 items with five types of response (1—not at all characteristic of me, 5—entirely characteristic of me), which evaluates the tendency to react negatively on an emotional, cognitive and behavioral level to uncertain situations and events. Two factors are presented: uncertainty to prospective intolerance and uncertainty to inhibitory uncertainty. It has been shown to have good psychometric properties. Cronbach’s alpha in this study was 0.91 to 0.94.

### 2.4. Data Analysis

Chi-square and one-factor ANOVA were used to examine potential group differences in clinical and demographic (age/gender) variables at the first evaluation. Two-factor repeated-measures ANOVA was conducted to test the effect of time (T1, T2) in the dependent variables in the groups of participants (OCD, SCZ, CG). Results of Greenhouse–Geisser correction were reported when Mauchly’s test of sphericity was significant. Statistically significant time by group interactions were followed up by Sidak’s post hoc comparisons. The partial Eta squared index was used to estimate the proportion of variance explained by time, interaction and group. In November, we evaluated intolerance to uncertainty and experiential avoidance, observing significant differences between groups, thus performing an analysis of covariance. Independent samples tests (Kruskal–Wallis H-test) were performed within each group, taking into account sex, education level and presence/absence COVID-19 patient. The Pearson correlation was used to analyze the relationship between variables age and coping strategies. SPSS Statistic 22.00 was used for statistical analysis.

## 3. Results

### 3.1. Changes in Coping Strategies, Anxiety and Depression

Results showed a significant intergroup, intragroup and interaction effect in anxiety, depression and coping strategies in both intergroups, except in intragroup avoidance (see Table 2).

The post hoc comparisons showed a significant effect in cognitive coping (T1) between CG–OCD and SCZ–OCD, the OCD group obtaining the worst result (*p* < 0.01). In T2, significant differences were between CG–OCD (*p* < 0.001). Social coping (T1) showed a significant effect between GG–SCZ (*p* < 0.001) and SCZ–OCD (*p* = 0.019). The SCZ group showed higher means than the other groups. These differences were maintained in T2. Avoidance coping (T1) reached significant differences between the OCD and SCZ groups versus CG (*p* < 0.001, *p* = 0.029, respectively), with remaining differences in November (*p* < 0.001, *p* = 0.008). In April, spiritual coping showed differences between OCD–CG (*p* = 0.005) and SCZ–CG (*p* < 0.001), the means being higher in the clinical groups. These differences were maintained in November, although the SCZ–OCD group reached new differences (*p* = 0.003), the means being higher in the SCZ group. Depression and anxiety in T1 were significantly different between CG-OCD (*p* <0.001) and SCZ–OCD (*p* < 0.01). In November, differences were maintained in anxiety between CG–OCD (*p* < 0.001) and SCZ–OCD (*p* = 0.001), while in depression, differences appeared between CG–SCZ (*p* <0.001). Intragroup comparisons only reported changes in CG in depression (*p* = 0.047) and in the SCZ group in cognitive coping (*p* < 0.001), social coping (*p* = 0.006), spiritual coping (*p* < 0.001), anxiety (*p* < 0.001) and depression (*p* < 0.001). OCD participants achieved significant changes in cognitive coping (*p* = 0.005), avoidance coping (*p* = 0.002), anxiety (*p* = 0.037) and depression (*p* < 0.001).

### 3.2. Coping Strategies Controlling Level of Intolerance to Uncertainty and Experiential Avoidance during Second Time

The groups presented statistically significant scores in experiential avoidance and uncertainty intolerance (see Table 3). Therefore, an analysis of covariance was performed.

Experiential avoidance affected cognitive coping (*p* = 0.025), maintaining differences between the CG–OCD group. Inhibitory uncertainty intolerance influenced social coping (*p* = 0.012), presenting the same differences as the uncontrolled SCZ–CG (*p* < 0.001) and SCZ–OCD (*p* = 0.001). Experiential avoidance and intolerance to inhibitory uncertainty influenced avoidance coping (*p* < 0.001), with no differences between groups. It did not affect any covariates to spiritual coping. The results are presented in Table 4.

### 3.3. Intragroup Comparison Based on Coping Strategies and Clinical and Sociodemographic

As for sex, differences were observed in social coping (*p* < 0.05) and spiritual coping (*p* < 0.001) in CG and OCD. The women achieved higher scores. The SCZ group showed no differences by sex.

Educational level in the CG presented differences in cognitive coping (*p* = 0.05), cognitive coping being greater in participants with university studies compared to those with primary education. Differences were also obtained in social coping (*p* < 0.001) and spiritual coping (*p* = 0.026), highlighting subjects with primary education. The OCD group with a primary education level had lower scores in cognitive coping (*p* = 0.014) and more in social coping (*p* = 0.009) than participants with other higher educational levels. Educational level did not influence the SCZ group.

Having suffered from COVID-19 influenced spiritual coping in CG (*p* = 0.028) and OCD (*p* = 0.001), with the average of those who suffered from this problem being higher. The OCD group also presented differences in avoidance coping in the same direction (*p* = 0.010).

In the CG, age was related to cognitive coping (*r* = 0.22; *p* < 0.001), social coping (*r* = −0.26, *p* < 0.001) and spiritual coping (*r* = −0.24, *p* < 0.001). In the OCD group, age was significantly correlated with cognitive coping (*r* = −0.21, *p* = 0.022). The SCZ group did not present significant relationships between age and coping strategies.

## 4. Discussion

The advance of COVID-19 and its consequences, together with the prevention behaviors used, might be increasing the risk of presenting symptoms and/or behavioral disorders such as anxiety disorders, depressive disorders, post-traumatic stress disorder, etc. One of the primary aims of this study was to examine the temporal change in coping strategies and levels of anxiety and depression in two groups of patients with prior psychopathologies (OCD and SCZ) versus a healthy CG during COVID-19. The evaluation moments were: 15–30 April 2020 (T1, first wave of COVID-19 in Spain) and 15–30 November 2020 (T2, second wave). Results indicated an increase in depressive responses in the three groups, while anxiety increased only in the clinical groups, highlighting the levels of the OCD group in both variables, perhaps as we are facing a disorder in which the elements of containment of the pandemic could produce a reinforcement of obsessions/compulsions, with the consequent increase in anxiety and depression [7,9,10].

Regarding coping strategies, we observed that the clinical groups decreased in coping cognitive—the SCZ group achieving the highest change score—and the more maladaptive strategies increased. Specifically, the OCD group increased above all coping avoidance (denial and self-blame) and coping social, which could be related to obsessions of guilt and verification. Over-seeking help, advice and information could exacerbate obsessive and anxious behaviors, in some cases becoming checking compulsions [13]. The SCZ group increased coping social and coping spiritual (seeking help, advice and information and turning to religion in times of stress or danger), something that could enhance paranoid beliefs [26,27]. The greater use of avoidance or blocking as a coping strategy in both clinical groups could be related both to the symptoms of the disorders themselves, and to the current situation, full of prevention and avoidance behaviors to avoid contagion.

Therefore, clinical groups increased the use of more maladaptive strategies than CG over time. This might indicate that the duration of COVID-19 not only produced changes at the level of anxiety and depression in clinical groups but also at the level of strategies to face this epidemic [21].

Our second aim was to analyze whether the coping strategies used in the second phase of the study (T2) could be influenced by experiential avoidance and intolerance to uncertainty. In these variables, the scores obtained by the clinical groups were higher than those of the control group, something expected by the type of population analyzed [15,26,27,28]. Coping avoidance was the strategy most affected by experiential avoidance and intolerance to inhibitory uncertainty, eliminating the differences between the clinical groups and the CG. People with a high score in experiential avoidance reject negative thoughts, feelings and situations to reduce the anguish that it generates, therefore being an avoidance strategy and influencing its results. As other authors have reported it could occur that experiential avoidance could have been increased by the stressful situation of the pandemic and be the most used response transforming the fear of COVID-19 into symptoms of OCD through dysfunctional avoidance behaviors owing to their cognitive errors, reinforcing obsessions and compulsions [21]. People with a high intolerance to inhibitory uncertainty present emotions and thoughts that prevent adaptive and healthy behavior, thus influencing avoidance strategies. That is, the more intolerant a person is to uncertainty, the more aversive they will find uncertainty in that situation, the more they will perceive the situation as uncertain and threatening. In situations with real uncertainty and real threat, this might increase worry and anxiety at the point we find ourselves in [14,25].

The third aim sought to verify whether sex, age, educational level and having suffered from COVID-19 could influence the use of one type of coping or another in each group. Regarding sex, it was found that the use of social and spiritual coping was higher in women in CG and OCD, coinciding with results of other authors, and thus, they may be more exposed to developing mental health problems during this pandemic [14]. Educational level only influenced the OCD and CG group, presenting a greater use of positive strategies in participants with higher education compared to those with primary education. The participants with a higher educational level might have been able to behave more actively in the face of the problem, plan and reinterpret it positively and, in turn, develop less aversive responses to a stressful situation. However, this variable could be related to the level of economic income or work situation, as in other research [3]. Regarding age, we could conclude that younger participants have used more dysfunctional strategies, coping worse with the pandemic, an aspect documented in previous studies [3,29]. Finally, suffering or having suffered from COVID-19 influenced the results of spiritual coping in CG and OCD, the average of those who suffered from this problem being higher. To embrace religion, believing that something can save us in times of illness could be a maladaptive strategy if it prevents us from facing the problem, or positive if, contrarily, it forces us to look for resources to overcome it. In this case, we consider it to be negative since it is associated with higher levels of anxiety.

From the results of this study, we can extract important implications for clinical practice. First, it is important to develop and implement effective screening procedures to identify risk and resilience factors and provide accurate intervention. We have observed that people with previous psychopathologies, with high experiential avoidance and intolerance to uncertainty, young people, women and those with a low educational level and who suffer/have suffered from COVID-19 present more unfit coping strategies with the consequent increase in higher levels of anxiety and depression. Training early in strategies to cope with stressful events could improve the mental health of the population. On the other hand, and because the pandemic could take time to disappear, psychoeducation is required during COVID-19; distinguishing between rational and adaptive rituals and compulsive acts, the control of exposure and consumption of information from media would help to control the fear level, particularly in the clinical population [21].

Among the limitations of this study, firstly, it is a study focused only on the Spanish population, with a non-random sample and with evaluation online; therefore, the administration conditions and situational variables that could influence their completion and responses could not be controlled. Another aspect to mention was the high mortality at the time of the second evaluation moment.

## 5. Conclusions

This study is the first to analyze the change at individual level during the first and second waves of the COVID-19 epidemic in Spain. The levels of anxiety and depression together with the maladaptive strategies increased in the clinical groups, while in the healthy group, only the depressive responses increased. We identified that the younger population, women, those with primary education levels and who suffered/had suffered the disease with greater experiential avoidance and intolerance to uncertainty could be risk factors for the use of dysfunctional coping strategies with the consequent deterioration of mental health.

## Figures and Tables

**Figure 1 brainsci-11-00926-f001:**
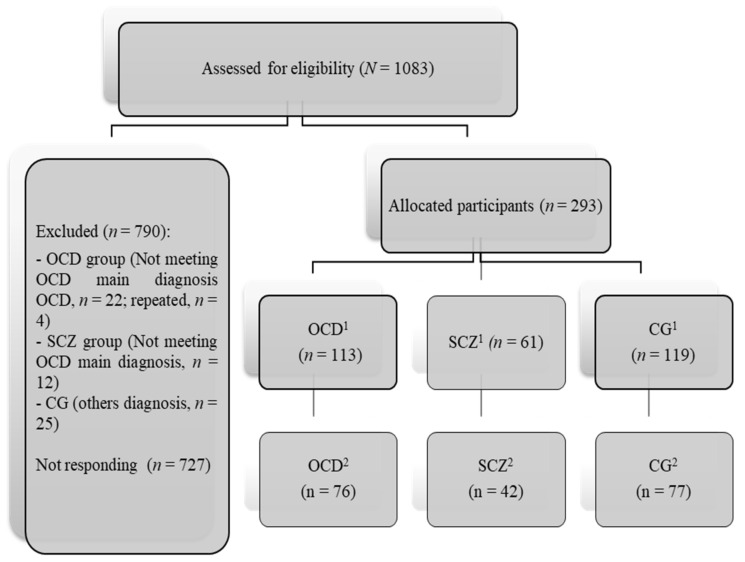
CONSORT Flow diagrams of study development. Note: OCD: obsessive–compulsive disorder, SCZ: schizophrenia; CGC: control group. ^1^ April; ^2^ November.

**Table 1 brainsci-11-00926-t001:** Sample characteristics T1.

Characteristics	OCD (*n* = 113)	SCZ (*n* = 61)	CG (*n* = 119)	ANOVA/χ^2^
Age (Mean ± SD)	34.54 ± 10.31	43.98 ± 10.59	35.05 ± 13.58	F (2;277,08) = 14,53; *p* < 0.001
Sex *n* (%)				Χ^2^ (2) = 25,57; *p* < 0.001
Men	61 (54)	21(34.4)	67 (56.3)	
Women	52 (46)	40 (65.6)	52 (43.7)	
Marital status *n* (%)				Χ^2^ (6) = 42,32; *p* <.001
Single	65 (57.5)	45 (73.7)	61 (51.2)	
Married	39 (34.5)	2 (3.3)	49 (41.1)	
Divorced	9 (8)	12 (19.7)	6 (5.2)	
Widower	0 (0)	2 (3.3)	3 (2.5)	
Educational level *n* (%)				Χ^2^ (6) = 10.95; *p* = 0.012
Elementary	6 (5.7)	32 (52.4)	2 (2)	
Secundary education	22 (19.5)	13 (21.3)	6 (5.2)	
High school	29 (25.2)	11 (18)	51 (24.4)	
University student	56 (49.6)	5 (8.3)	60 (72.3)	
Who did you spend quarantine with *n* (%)				Χ^2^ (2) = 10.95; *p* = 0.012
Alone	9 (7.9)	13 (21.3)	9 (7.6)	
Accompanied	104 (92.1)	48 (78.9)	110 (92.4)	
COVID-19 patient				ns
Yes	5 (4.8)	4 (6.6)	5 (4)	
No	108 (95.2)	57 (93.4)	114 (96)	

T1: 15–30 April 2020; OCD: obsessive–compulsive disorder; SCZ: schizophrenia; CG: control group; *n* = number; SD = standard deviation; ns: not significant.

**Table 2 brainsci-11-00926-t002:** ANOVA mixed in coping strategies, anxiety and depression.

Time	OCD	SCZ	CG	F		*p*	η^2^
	(n_april_ = 113; n_november_ = 85)	(n_april_ = 61; n_november_ = 42)	(n_abril_ = 119; n_noviembre_ = 77)				
	*M ± SD*	*M ± SD*	*M ± SD*				
Cognitive C.				F(time)=	44.27	<0.001	0.1
T1	16.48 ± 5.76	19.68 ± 6.24	18.96 ± 5.01	F(interaction)=	19.62	<0.001	0.1
T2	16.02 ± 5.32	17.65 ± 5.54	18.98 ± 4.97	F(group)=	11.59	<0.001	0.06
Social C.				F(time)=	10.99	0.001	0.03
T1	8.95 ± 4.49	10.88 ± 3.66	8.00 ± 3.95	F(interaction)=	3.09	0.047	0.02
T2	9.09 ± 4.41	11.29 ± 3.84	8.03 ± 3.92	F(group)=	9.58	<0.001	0.05
Avoidance C.				F(time)=	2.28	>0.05	0.01
T1	8.68 ± 3.86	8.34 ± 3.18	6.68 ± 3.23	F(interaction)=	5.65	0.004	0.03
T2	8.96 ± 3.77	8.45 ± 3.01	6.59 ± 3.09	F(group)=	19.07	<0.001	0.1
Spiritual C.				F(time)=	35.57	<0.001	0.08
T1	1.93 ± 1.98	2.69 ± 2.36	1.28 ± 1.69	F(interaction)=	23.81	<0.001	0.11
T2	1.95 ± 1.97	3.12 ± 2.19	1.27 ± 1.66	F(group)=	14.65	<0.001	0.07
Anxiety				F(time)=	24.41	<0.001	0.06
T1	11.69 ± 4.45	8.41 ± 4.81	7.44 ± 3.76	F(interaction)=	6.96	0.001	0.03
T2	11.85 ± 4.22	9.06 ± 4.35	7.51 ± 3.69	F(group)=	48.13	<0.001	0.19
Depression				F(time)=	102.98	<0.001	0.2
T1	8.09 ± 4.27	5.52 ± 4.08	5.00 ± 3.57	F(interaction)=	31.33	<0.001	0.14
T2	8.61 ± 3.94	7.67 ± 2.34	5.17 ± 3.52	F(group)=	33.03	<0.001	0.14

OCD: obsessive–compulsive disorder; SCZ: schizophrenia; CG: control group. *M* = mean; *SD* = standard deviation; C.: coping; T1: 15–30 April 2020; T2: 15–30 November 2020.

**Table 3 brainsci-11-00926-t003:** ANOVA in experiential avoidance and uncertainty intolerance.

Variables	OCD (*n* = 76)	SCZ (*n* = 42)	CG (*n* = 77)	F
Experiential avoidance				F = 22.55; *p* < 0.001
(*Mean ± SD*)	56.47 ± 13.92	57.12 ± 12.95	48.14 ± 11.47	
Uncertainty intolerance				F = 37.22; *p* < 0.001
(*Mean ± SD*)	82.60 ± 19.24	74.75 ± 23.70	62.28 ± 21.21	
Uncertainty to prospective intolerance				F = 14.29; *p* < 0.001
(*Mean ± SD*)	31.61 ± 8.93	30.62 ± 10.14	26.23 ± 9.30	
Uncertainty to inhibitory uncertainty				F = 54.06; *p* < 0.001
(*Mean ± SD*)	51.00 ± 11.57	44.12 ± 14.42	35.93 ± 13.31	

OCD: obsessive–compulsive disorder; SCZ: schizophrenia; CG: control group; *n* = number; *SD* = standard deviation.

**Table 4 brainsci-11-00926-t004:** ANCOVA coping strategies with experiential avoidance and uncertainty intolerance.

Variables	OCD (*n* = 113) M Adjusted	SCZ (*n* = 61) M Adjusted	CG (*n* = 119) M Adjusted	F	*p*
Cognitive coping	16.48	18.72	18.70	6.03	0.003
Social coping	8.53	11.81	8.37	7.95	<0.001
Avoidance coping	7.89	8.19	7.14	2.99	0.05
Spiritual coping	1.93	3.10	1.26	13,04	<0.001

OCD: obsessive–compulsive disorder; SCZ: schizophrenia; CG: control group; M adjusted = mean adjusted; *SD* = standard deviation.

## Data Availability

The data that support the findings of this study are available on request from the corresponding author.

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
