# Peer review of "Coping Strategies, Anxiety and Depression in OCD and Schizophrenia: Changes during COVID-19"

_brainsci, 2021, doi:10.3390/brainsci11070926_

Round 1
Reviewer 1 Report
The authors are trying to answer an interesting question which is whether the use of coping strategies in clinical populations change with progression of the pandemic and whether this is potentially reflected in the or linked to mental health outcomes, such as anxiety and depression, but also avoidance and tolerance of uncertainty.
The idea, as already said, is interesting however I am doubtful about the soundness of the statistical procedures. In fact a major revision of the analysis section and the presentation of the results is required. At the moment I don't believe the interpretations drawn from the tests/results presented.
One additional concern is that no comment about patients medication has been made, which is highly problematic, as e.g. most chronic schizophrenia patients will receive antidepressants, which will strongly impact the results.
Please see comment in order of appearance:
Abstract:
Divide N by group for second timepoint
Introduction:
What is the difference between the prevention strategy: “isolation” and the individual variables: “social isolation”? Previous psychopathology is a very clear individual variable.
Last sentence of the intro: The third objective sought to verify whether sex, age, educational level and whether or not having suffered COVID-19 could influence the use of one type of coping or another in group. Word missing “having suffered FROM A COVID-19 INFECTION could influence”
A stronger motivation for this study is required, it is difficult to link the three objectives - especially the last one seems out of context/ motivation
Participants:
Please provide Ns for the second timepoint and not percentages? Were patients, especially schizophrenia patients medicated for their negative symptoms, especially with antidepressants? SCZ patients seem pretty low on their T1 Depression score - it would be essential to control for the use of antidepressant medication, is that the reason why the initial score is so low? How are those with and without antidepressant medication behaving.
Also, showing individual data would be extremely informative. I’d suggest the authors include box plots for their outcome variables which shows the development of the individuals with lines (for those who participated twice)
Procedures:
The authors state that “all families provide written informed consent”, does that mean that not all patients were able to provide informed consent? Please clarify!
Data analysis:
The authors are stating the following: Two-factor repeated-measures ANOVA was conducted to test the effect of the treatments as a function of time (T1, T2) and group (OCD, SCZ, CG). What is the treatment the authors are referring to, I guess it is the coping strategies and mental health outcomes? That also means that the final statistics is being run on only those subjects that have taken part in both timepoints. There it is incorrect to state that all subjects have been used for analysis. The authors should consider presenting everything only on those individuals with both timepoints. The authors should be careful to use terminology that is correct and makes sense in their setting:
Results of Greenhouse-Geisser correction were reported Mauchly’s test of sphericity was significant. Fullstop missing.
Statistically significant time by condition interactions were followed up by Sidak’s post-hoc comparisons. What is the condition the authors are referring to? The group or what they previously referred to as treatment?
The partial Eta squared index was used to estimate the proportion of variance explained by each source. What is the source they are referring to?
Table 2 should be reformatted as timepoint is listed under DV – dependent variable, but the timepoint is independent and should be listed under group – or simply rephrase the DV.
Those are simple post-hoc test and should be described as such.
What is the motivation for the intragroup comparisons?
Considering the standard deviation and the results presented in table 2 – it doesn’t seem true that the three groups increase significantly in depression and anxiety, this result must be driving by the scz-group, what postdoc test did the authors use? The tests reported show only differences between groups, but not time-points within groups. Therefore the result summary at the beginning of the discussion is not correct. The same holds for the coping strategies.
Author Response
Dear reviewer 1
First of all, thank you very much for the suggested contributions to this study as this allows us to make improvements. We appreciate your invaluable time.
We have modified the areas suggested.
Faithfully,
Ana Isabel Rosa-Alcázar
Reviewer
The authors are trying to answer an interesting question which is whether the use of coping strategies in clinical populations change with progression of the pandemic and whether this is potentially reflected in the or linked to mental health outcomes, such as anxiety and depression, but also avoidance and tolerance of uncertainty.
The idea, as already said, is interesting however I am doubtful about the soundness of the statistical procedures. In fact a major revision of the analysis section and the presentation of the results is required. At the moment I don't believe the interpretations drawn from the tests/results presented.
One additional concern is that no comment about patients medication has been made, which is highly problematic, as e.g. most chronic schizophrenia patients will receive antidepressants, which will strongly impact the results.
Please see comment in order of appearance:
Reviewer: Abstract: Divide N by group for second timepoint
Authors: The information requested is included in Abstract.
“comprised 195 participants (85 Obsessive-Compulsive patients, 42 schizophric patients and 77 healthy controls)”
Reviewer: Introduction: What is the difference between the prevention strategy: “isolation” and the individual variables: “social isolation”? Previous psychopathology is a very clear individual variable.
Authors:
The authors agree with the review and know the importance of social isolation as a risk factor in the development of psychopathologies. In the introduction we have tried to differentiate between stressors of the COVID-19 disease itself (duration, type, number, anger, ability to control, etc.) and the preventive behaviors necessary to prevent contagion (social distancing, hand washing, use of a mask, among others) of risk factors for mental health, among which loneliness, social assimilation, level of fear, etc. We have removed isolation from the manuscript in order to avoid confusion and we have included the following paragraphs in the manuscript
“Social isolation is a risk factor for the mental disorders, being a mediating factor for depression, anxiety, suicidal ideation and psychotic episodes [11,12]
Reviewer: Last sentence of the intro: The third objective sought to verify whether sex, age, educational level and whether or not having suffered COVID-19 could influence the use of one type of coping or another in group. Word missing “having suffered FROM A COVID-19 INFECTION could influence”
Authors: The authors agree with the reviewer and modify what was suggested
Reviewer: A stronger motivation for this study is required, it is difficult to link the three objectives - especially the last one seems out of context/ motivation.
Authors: The authors have included the following paragraphs in the manuscript.
Focusing on OCD, the elements of containment of the pandemic will produce reinforcement of obsessive thinking and compulsive behavior and exacerbation of symptoms [7,9,10]. Others authors reported a higher perception of risk, fear, anxiety and stress in a schizophrenic population during a swine fever pandemic [8].
Sex, age, educational level have been predictors or mediators of mental health during COVID-19 [3]. Similarly, having suffered from the psychopathological disorder or having relatives with an infected member also influenced an increase in stress, anxiety and depression [22]
Reviewer: Participants: Please provide Ns for the second timepoint and not percentages?
Authors: The information requested is included in Participants
85 obsessive-Compulsive patients, 42 schizophrenic patiens, and 77 healthy controls participants.
Reviewer: Were patients, especially schizophrenia patients medicated for their negative symptoms, especially with antidepressants? SCZ patients seem pretty low on their T1 Depression score - it would be essential to control for the use of antidepressant medication, is that the reason why the initial score is so low? How are those with and without antidepressant medication behaving.
Authors: Both obsessive-compulsive and schizophrenic patients were taking antidepressant and, in some cases, antipsychotic medication. We include this information in the manuscript.
The clinical participants were medicated with antidepressants and, in some cases, antipsychotics.
Reviewer: Also, showing individual data would be extremely informative. I’d suggest the authors include box plots for their outcome variables which shows the development of the individuals with lines (for those who participated twice)
Authors: The authors present box plots in supplementary file 1
Reviewer: Procedures: The authors state that “all families provide written informed consent”, does that mean that not all patients were able to provide informed consent? Please clarify!
Authors: The authors clarify the information and modified the manuscript.
All participants or their tutors provide written informed consent
Reviewer: Data analysis: The authors are stating the following: Two-factor repeated-measures ANOVA was conducted to test the effect of the treatments as a function of time (T1, T2) and group (OCD, SCZ, CG). What is the treatment the authors are referring to, I guess it is the coping strategies and mental health outcomes?
Authors: The authors indicate that it is a transcription error and correct the manuscript.
Two-factor repeated-measures ANOVA was conducted to test the effect of time (T1, T2) in the dependent variables in the groups of participants (OCD, SCZ, CG).
Reviewer: That also means that the final statistics is being run on only those subjects that have taken part in both timepoints. There it is incorrect to state that all subjects have been used for analysis. The authors should consider presenting everything only on those individuals with both timepoints. The authors should be careful to use terminology that is correct and makes sense in their setting:
Authors: As the reviewer indicates, we delete that paragraph. The authors consider that it is important not to lose subjects, therefore, we maintain the analysis with the complete sample.
Reviewer: Results of Greenhouse-Geisser correction were reported Mauchly’s test of sphericity was significant. Fullstop missing.
Authors: The authors corrected the error
Results of Greenhouse-Geisser correction were reported when Mauchly’s test of sphericity was significant
Reviewer: Statistically significant time by condition interactions were followed up by Sidak’s post-hoc comparisons. What is the condition the authors are referring to? The group or what they previously referred to as treatment?
Authors: The authors corrected the error.
Statistically significant time by group interactions were followed up by Sidak’s post-hoc comparisons
Reviewer: The partial Eta squared index was used to estimate the proportion of variance explained by each source. What is the source they are referring to?
Authors: The authors modify the paragraph
The partial Eta squared index was used to estimate the proportion of variance explained by time, interaction and group
Reviewer: Table 2 should be reformatted as timepoint is listed under DV – dependent variable, but the timepoint is independent and should be listed under group – or simply rephrase the DV.
Authors: The authors have decided to remove DV and include Time
Reviewer: Those are simple post-hoc test and should be described as such.
Authors: The authors have included post-hoc comparisons
Reviewer: What is the motivation for the intragroup comparisons?
Authors: The motivation of the intragroup comparisons is to analyze whether some sociodemographic variables or the psychopathological condition could influence each group of participants. Previous studies reported predictors that could influence the discomfort caused by COVID-19. We have tried to check if this occurred in each of the groups of participants, thus being able to take into account not only psychopathology but other relevant variables (Maguire, Reay and Looi, 2019; Mazza, 2020, etc.)
Reviewer: Considering the standard deviation and the results presented in table 2 – it doesn’t seem true that the three groups increase significantly in depression and anxiety, this result must be driving by the scz-group, what postdoc test did the authors use? The tests reported show only differences between groups, but not time-points within groups. Therefore the result summary at the beginning of the discussion is not correct. The same holds for the coping strategies.
Authors: The information provided is based on the intragroup comparisons described in the text of the results.
Reviewer 2 Report
In the introduction, please better justify the selection of such clinical groups for the study.
The statistical analysis was done in an understandable way and focuses on the main idea of the article. However, I have a few specific comments:
Please prepare a longer description of the Brief Experiential Avoidance Questionnaire
Please report the value of the F-statistic in Table 2.
In the description of the interaction effect, please indicate the F value for simple effects
Please prepare a plot for table 2, which would show graphically the differences obtained when comparing the groups.
Author Response
Reviewer 2
Dear reviewer 2,
First of all, thank you very much for the suggested contributions to this study as this allows us to make improvements. We appreciate your invaluable time.
We have modified the areas suggested.
Faithfully,
Ana Isabel Rosa-Alcázar
Reviewer: In the introduction, please better justify the selection of such clinical groups for the study.
Authors: The authors have included the following paragraphs in the manuscript.
Focusing on OCD, the elements of containment of the pandemic will produce reinforcement of obsessive thinking and compulsive behavior and exacerbation of symptoms [7,9,10]. Others authors reported a higher perception of risk, fear, anxiety and stress in a schizophrenic population during a swine fever pandemic [8].
Reviewer: The statistical analysis was done in an understandable way and focuses on the main idea of the article. However, I have a few specific comments:
Please prepare a longer description of the Brief Experiential Avoidance Questionnaire
Authors: The authors include the following paragraph
Self-report measure of experiential evitation of 15 items rated a 6 point Likert scale (1 to 6). Higher scores indicate greater experiential avoidance.
Reviewer: Please report the value of the F-statistic in Table 2.
Authors: The authors include the information requested by the reviewer
Reviewer: In the description of the interaction effect, please indicate the F value for simple effects
Authors: The authors include the information requested by the reviewer
Reviewer: Please prepare a plot for table 2, which would show graphically the differences obtained when comparing the groups.
Authors: The authors include 6 graphs in supplementary file 2.